# Persistent Homology with Improved Locality Information for more Effective Delineation

## Abstract

We present a new, more effective way to use Persistent Homology (PH), a method to compare the topology of two data sets, for training deep networks to delineate road networks in aerial images and neuronal processes in microscopy scans. Its essence is in a novel filtration function, derived from a fusion of two existing techniques: thresholding-based filtration, previously used to train deep networks to segment medical images, and filtration with height functions, used before for comparison of 2D and 3D shapes. We experimentally demonstrate that deep networks trained with our Persistent-Homology-based loss yield reconstructions of road networks and neuronal processes that preserve the connectivity of the originals better than existing topological and non-topological loss functions.

## 1 Introduction

In many image segmentation tasks, the topology of the resulting mask is as important as, if not more than, its pixel-wise accuracy. For example, a model of an aortic valve that does not form a ring is biologically implausible. Similarly, networks of curvilinear structures—-be they roads in aerial images, blood vessels in Computer Tomography (CT) scans, or dendrites and axons in Light Microscopy (LM) image stacks—should not feature breaks that disrupt connectivity or false connections between disjoint structures. Unfortunately, deep networks trained by minimizing pixel-wise loss functions, such as the cross-entropy or the mean square error, are subject to such mistakes. This is in part because it often takes very few mislabeled pixels to alter the topology significantly with little impact on the pixel-wise accuracy. In other words, it is possible for a network trained in this manner to deliver both a good pixel classification accuracy and an incorrect topology.

Specialized solutions to this problem have been proposed in the form of loss functions that compare the topology of the prediction to that of the annotation. They are effective for specific applications but do not naturally generalize. For example, the perceptual loss of (Mosińska et al., 2018) penalizes topological differences between the prediction and the ground truth, but cannot be guaranteed to detect them all. Similarly, minimizing the MALIS loss for segmenting electron microscopy scans (Briggman et al., 2009; Funke et al., 2018) yields better region boundaries but does not penalize interruptions in loopy linear structures. This has been addressed by (Oner et al., 2021) for delineation of 2D road networks but the proposed solution is not applicable to 3D image stacks.

Persistent Homology (PH) (Edelsbrunner & Harer, 2008) is an elegant approach to describing and comparing topological structure of data, which is well-established in the field of topological data analysis (TDA). It offers the promise to address the connectivity problem in a generic way, both for 2D and 3D images. Homology is the study of topological features in an object, such as its connected components (0-homology classes), loops (1-homology classes) and closed surfaces (2-homology classes). Persistent homology detects homology classes in objects *filtered* at different *scales*. A homology class which appears at a particular scale and disappear at a larger one is represented by a scale interval called a *persistence interval*. The set of persistence intervals for all the homology classes characterizes the overall topology of the structure. It can be represented by a *persistence diagram*. The similarity of these diagrams across two different structures can then be used to quantify their topological similarity. This has been successfully exploited to train deep networks for delineation (Hu et al., 2019), image segmentation (Hu et al., 2019; Clough et al., 2019; 2020) and crowd counting (Abousamra et al., 2021).

However, existing techniques do not unleash the full power of persistent homology because the persistent diagrams are global image descriptors that ignore the location of the topological features,

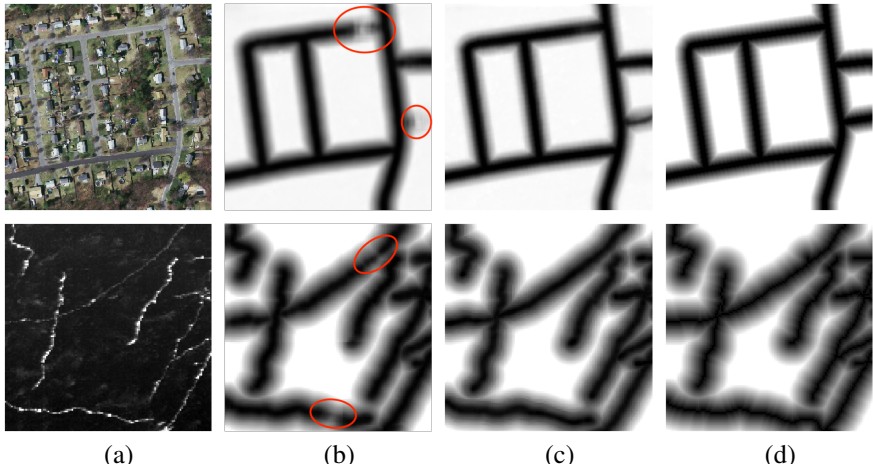

|  (a)  |  (b)  |  (c)  |  (d)  |

Figure 1: **2D and 3D delineation.** (a) Aerial image and slice of a microscopy stack. (b) A network trained using a standard homology-based loss yields road and neurite interruptions. (c) One trained using our localized loss is more topologically accurate and produces predictions that closely resemble the ground truth (d).

which reduces their descriptive power. As shown in Fig. 1, this can result in networks that still fail to enforce the proper topology. This is because, when training a deep network, a persistence-based loss can be low even if the network predicts a structure that is quite different from the ground-truth. To remedy this, we introduce a new approach to computing persistence diagrams that takes location into account and increases their descriptive power, as shown in Fig. 3. Our main contribution is a novel filtration technique that combines two filtrations commonly used in TDA: thresholding based filtrations and height functions. It results in a loss that applies both to 2D and 3D images and significantly improves performance compared to state-of-the-art topological methods.

## 2 RELATED WORK

Training a deep network that produces topologically correct segmentations has typically been done by designing loss functions that, when minimized, favor plausible topology. In this section, we briefly review first those that do not rely on Persistent Homology, and then those that do.

**Losses designed to enforce topological correctness** Several such losses have been proposed already to go beyond pixel-wise classification accuracy by encoding more global properties. In (Li et al., 2020), the connectivity between neighboring pixel pairs is used as an additional source of supervision. This approach has been shown to improve connectivity, but since disconnections or false connections are not penalized explicitly, there is no guarantee it captures all such errors. The perceptual loss of (Mosińska et al., 2018) is based on the assumption that a pre-trained neural network can capture differences of connectivity between the prediction and the ground-truth. However, even though it has been shown experimentally to improve the topology of masks produced by a deep net, there is no guarantee that this assumption holds in general. Making the Rand index of segmentations produced by the network similar to that of ground truth ones (Briggman et al., 2009; Funke et al., 2018) helps when modeling tree-like structures, both in 2D and in 3D, but cannot prevent disconnections in loopy structures. This shortcoming has been addressed by (Oner et al., 2021) by detecting disconnections of 2D loopy structures as interconnections of background regions, but the proposed solution does not generalize to 3D.

**Losses that rely on Persistent Homology** Persistent Homology (Edelsbrunner et al., 2000; Zomorodian & Carlsson, 2004) is an established topological data descriptor. Among its numerous applications is comparing topological structures of binary images. It has been used to enforce the correct Betti number on binary masks resulting from inference in Markov Random Fields (Chen et al., 2011). Recently, it has been demonstrated that persistence diagrams can be computed also for greyscale images and differentiated with respect to the pixel values (Hu et al., 2019; Clough et al., 2019; Gabrielsson et al., 2020; Leygonie et al., 2021; Carriere et al., 2021). Hence, they can be used as loss or prior terms for training deep networks. In this vein, (Clough et al., 2019) proposed a loss term that enforces a sequence of desired Betti numbers on the predicted segmentation. This approach was further extended to a loss function that tends to equalize the Betti number of the prediction and the ground truth (Clough et al., 2020). (Hu et al., 2019) proposed instead to construct a

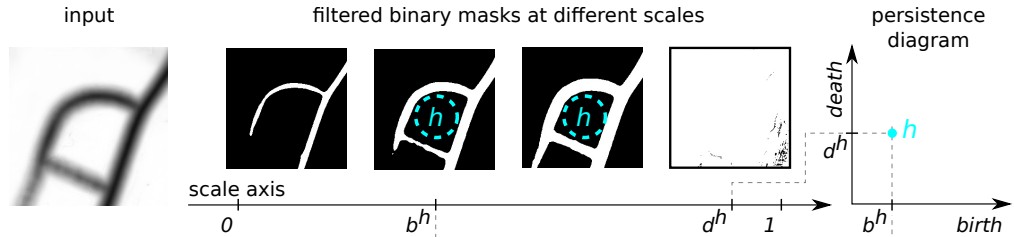

Figure 2: **Filtration.** When the distance map shown on the left is filtered by thresholding, the loop $h$ emerges at scale $b^h$ and is filled at scale $d^h$. This gives rise to the point $(b^h, d^h)$ in the persistence diagram shown on the right. Here, thresholding means retaining all pixels whose value is lower than the threshold.

loss term around a difference of persistence diagrams of the prediction and the ground truth. Their persistence diagrams are obtained by thresholding the prediction and the ground truth. As we show in the next section, for binary ground truth this results in trivial persistence diagrams, that encode no more information than the ground truth Betti number. In consequence, both approaches can be interpreted as equalizing the Betti numbers of the prediction and the ground truth. (Wang et al., 2020) improved upon this technique by applying it to predicted and ground truth distance maps instead of binary annotations and class affinity maps. As discussed in more detail in section 3, this makes the loss function more effective at detecting and penalizing topological errors. Unfortunately, even the improved technique remains susceptible to errors provoked by incorrectly matching the persistence diagrams of the prediction and the ground truth. By making the diagrams depend on location of the topological features in the image, our method makes them more diverse and minimizes potential for erroneous matches.

It has also been proposed to detect disconnections in predicted 2D and 3D structures using Discrete Morse Theory (Hu et al., 2021). Topological features that are inconsistent with the ground truth are then penalized in the loss function. However, when the annotations lack spatial precision, which is often the case for neurite and road centerline annotation like the ones studied here, ground-truth inaccuracies may confuse the network. By contrast, our technique allows for considerable misalignment between the prediction and the ground truth.

## 3 METHOD

We first introduce Persistent Homology and its application to characterizing two-dimensional images and three-dimensional image stacks. As PH provides global descriptors that ignore location of topological features, we then introduce our approach to accounting for it.

### 3.1 PERSISTENT HOMOLOGY

In the interest of simplicity, we introduce PH for binary images and image stacks, where homology classes are limited to connected components, loops, and closed surfaces. We refer the interested reader to the review (Edelsbrunner et al., 2000) for a more general treatment, applicable to non-image and higher-dimensional data.

At the heart of persistent homology is detecting homology classes (connected components, loops, closed surfaces) at many different scales. The ones that exist over a wide scale range are called persistent and deemed more likely to represent true features, as opposed to sampling artifacts or noise. Here, scale has a very specific meaning. It refers to the parameter of a filtration function $F$ that is applied to an image $X$ to produce topological objects called *cubical complex*. Cubical complexes resulting from filtering images and their properties are described for instance by (Garin et al., 2020) and by (Bleile et al., 2021). A reader not familiar with algebraic topology can think of them as binary masks. The masks obtained for different scales form a sequence of inclusions, that is, for a pair of scale parameters $s_1 < s_2$, the mask $F(\mathbf{X}, s_1)$ is entirely contained within the mask $F(\mathbf{X}, s_2)$. The simplest example of a function for filtering grayscale images is thresholding, where the threshold acts as the scale, as shown in Fig. 2.

As the scale changes, homology classes in the filtered cubical complex emerge and disappear. To capture this, the scale range is sampled from small to large, the image is filtered at the selected scale values, homology classes in the resulting binary masks are detected algebraically (Edelsbrunner et al., 2000), and correspondence is established between the homology classes found at consecutive

scales. For each class, this yields a pair $(b, d)$, where $b$ is the scale at which the homology class appears and $d$ the scale at which it disappears. We will refer to them as *birth* and *death* times and to the interval $[b, d]$ as the *persistence interval* of the homology class. The set $P_{\mathbf{X}} = \{(b^h, d^h)\}_{h \in H_{\mathbf{X}}}$, where $H_{\mathbf{X}}$ is the set of all homology classes found in the filtered image $\mathbf{X}$, is called the *persistence diagram* of $\mathbf{X}$, and was first introduced by (Barannikov, 1994). In practice, we use the Gudhi library (Maria et al., 2014) to compute persistence diagrams from images. Fig. 2 depicts the birth and death of a specific homology class.

To compare images $\mathbf{X}_1$ and $\mathbf{X}_2$, one-to-one matching is performed between their persistence diagrams, $P_{\mathbf{X}_1}$ and $P_{\mathbf{X}_2}$, with the cost of matching a homology $g \in H_{\mathbf{X}_1}$ to a homology $h \in H_{\mathbf{X}_2}$ set to $c_{g,h} = (b^h - b^g)^2 + (d^h - d^g)^2$ and the cost of leaving an interval $[b, d)$ unmatched is set to the distance between the point $(b, d)$ and the diagonal in $\mathbb{R}^2$. The optimal matching can be found using the Hungarian algorithm. Its cost that we denote as $C(\mathbf{X}_1, \mathbf{X}_2)$ quantifies the topological discrepancy between $\mathbf{X}_1$ and $\mathbf{X}_2$ by penalizing differences between corresponding homology classes and ones that only appear in either $\mathbf{X}_1$ or $\mathbf{X}_2$.

## 3.2 TRAINING DEEP NETWORKS USING PH

Let $f$ be a network that associates to an image $\mathbf{X}$ a segmentation mask $\mathbf{Y} = f(\mathbf{X})$ such that for all pixels or voxels $p \in \mathbf{Y}$, $0 \le \mathbf{Y}[p] \le 1$ and let $\hat{\mathbf{Y}}$ be the corresponding ground-truth mask. A natural idea then is to train $f$ by minimizing

$$L_{\text{tot}}(\mathbf{Y}, \hat{\mathbf{Y}}) = L(\mathbf{Y}, \hat{\mathbf{Y}}) + \alpha C(\mathbf{Y}, \hat{\mathbf{Y}}) , \tag{1}$$

where $L$ is the standard loss function, either the Mean Square Error, or the Cross Entropy, and $\alpha$ is a hyper-parameter, which we set to 0.01 in practice. This is possible because $C$ is sub-differentiable with respect to its inputs when filtration is achieved by thresholding, as shown before (Hu et al., 2019; Clough et al., 2019; Leygonie et al., 2021). However, when the ground truth $\hat{\mathbf{Y}}$ is binary, as it often is, all structures emerge at scale zero and disappear at scale one. Hence, as shown in Fig. 3(a) the persistence intervals all are $[0, 1]$ and filtering it by thresholding is uninformative. An approach to handling this difficulty is to replace the binary ground truth by its distance transform that can be thresholded over a wide range of threshold values to create different binary masks (Wang et al., 2020). Unfortunately, computing the persistence diagram of a ground truth distance transform still yields persistence diagrams in which the topological features of the original, binary ground truth are spread along the 'death' axis but not along the 'birth' one: The distance value at the structures themselves is zero and, as a result, all the loops of the ground truth mask appear as soon as the scale value becomes positive. As shown in Fig. 3(b), this may lead to erroneous matches between the persistence diagrams, which encourages the deep network to produce wrong segmentations. Moreover, this approach ignores the location of homology classes within the image. This is sub-optimal, because the predicted topological features should not be too far from the ground truth ones.

## 3.3 FILTRATION THAT PARTLY LOCATES TOPOLOGICAL FEATURES

To remedy the above-mentioned drawbacks of traditional PH, our goal is therefore to spread the persistence diagrams along *both* dimensions while also accounting for where in the image the homology classes are. To this end, we draw our inspiration from another filtration technique called the height function (Turner et al., 2014). It was originally designed for three-dimensional meshes and can be applied to binary images by assigning to each pixel a *height* value that is the coordinate of its projection along a selected straight line. Filtration is carried out by forming binary masks made of pixels whose height is smaller than the scale parameter (Garin & Tauzin, 2019). As the scale is increased, the binary image is revealed in scan-lines perpendicular to the height axis, one scan-line at a time. The birth and death times are the heights of pixels responsible for the emergence and disappearing of homology classes. As a result, the persistence diagram contains partial information about the location of topological features. Moreover, both birth and death times of different homology classes are distributed across scales. Additionally, it has been shown that a binary image can be reconstructed from as few as four persistence diagrams obtained with height functions with well-chosen directions (Betthauser, 2018). A height function is only defined for binary images, but the abovementioned result inspired us to extend its definition by combining it with thresholding distance maps. Given a scale $s$, the value of the filtered binary mask at coordinates $\mathbf{p}$ is taken to be

$$F(\mathbf{Y}, s)[\mathbf{p}] = \mathbb{1}(\mathbf{Y}[\mathbf{p}] + g(\mathbf{p}) < s) , \tag{2}$$

where $\mathbb{1}(\cdot)$ evaluates to one if the condition in the bracket is satisfied and to zero otherwise. In essence, this amounts to thresholding the sum of the height function $g$ and the pixel values. From

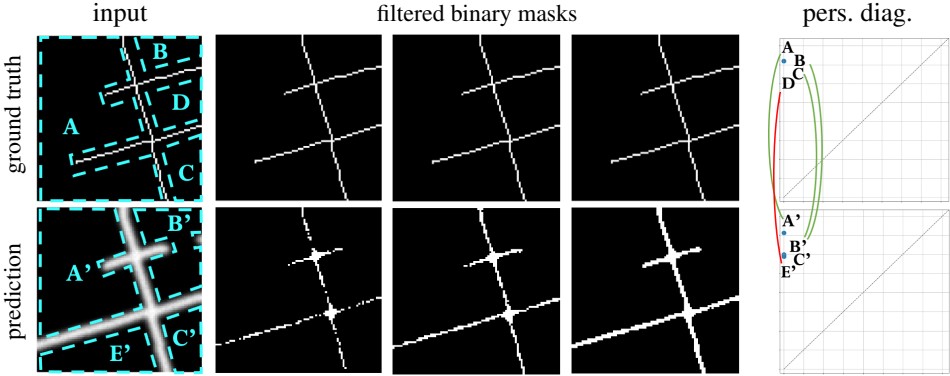

(a) **Filtration by thresholding** binary ground truth and predicted class affinity maps. Here, filtration involves decreasing the threshold from 1 to 0, and retaining the pixels greater than the threshold. Note, that the the binary masks resulting from filtering the ground truth at different scales are all the same and that all points in the ground truth persistence diagram (*top-right*) coincide. This results in erroneous matches between the predicted and ground truth homology classes. Minimizing a loss function based on such a filtration can magnify the errors.

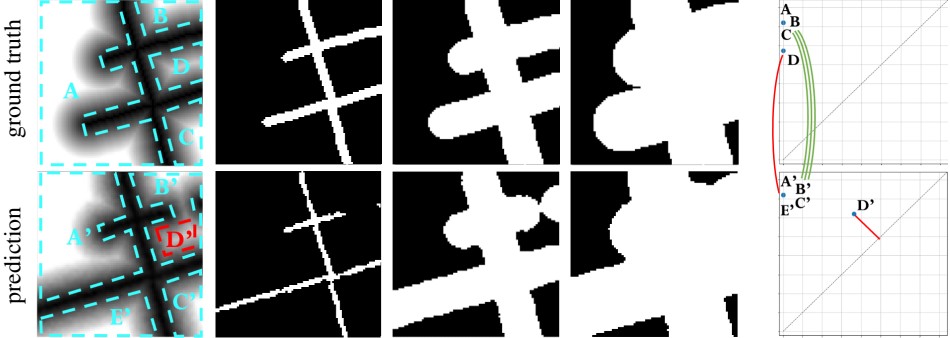

(b) **Filtration by thresholding distance maps** distributes the topological features of the ground truth along the vertical but not the horizontal axis. This still results in erroneous matching between the predicted and ground truth homology classes: Loop D' in the prediction emerges when the threshold is high enough to make the road brake disappear. Hence, it remains unmatched and the E' loop created by the false positive road is matched to the ground truth loop D.

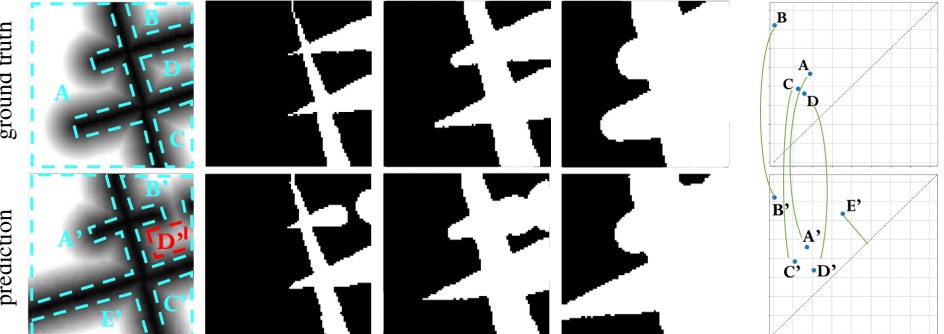

(c) **Our localized filtration of distance maps** distributes the persistence diagram of the ground truth across the plane, promoting correct matches between predicted and ground truth homology classes.

Figure 3: **Comparing filtration functions on synthetic data.** The binary ground truth road annotation (*top-left* in each table part) contains four loops, marked with cyan dashed lines. We synthesized a predicted class affinity map (*bottom-left* in each part) by extending one road to the left and interrupting another. In consequence, loop B and D from the ground truth are joined into B' in the prediction, and A is split into A' and E'. For each filtration method, we show binary masks resulting from filtration at different scales, pairs of persistence diagrams, and their optimal matches.

the perspective of TDA, such combination of two filtration functions can be seen as a line in the fibered barcode defined by (Carrière & Blumberg, 2020).

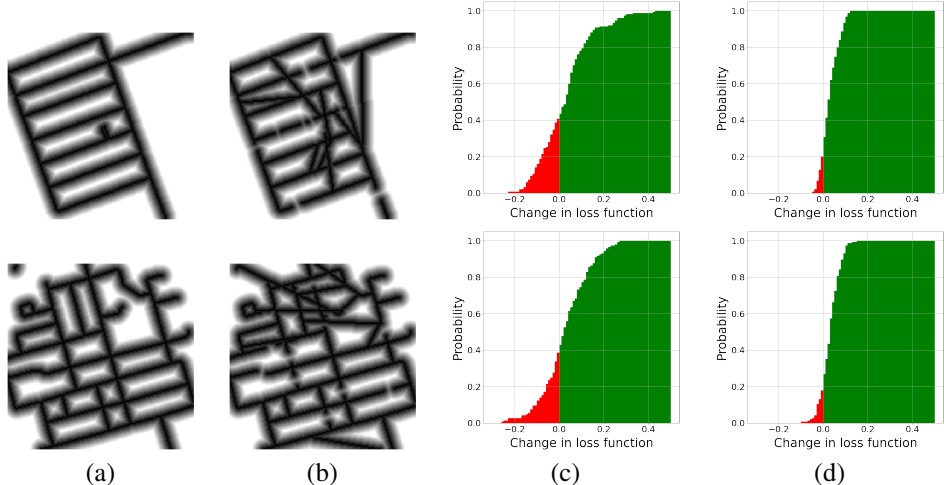

Figure 4: **Sensitivity of the topological loss term** $C$ **to the number of injected errors** (a) Ground truth distance maps of road networks. (b) Distance maps corrupted by introducing false roads and interruptions. We randomly injected one error at a time, obtaining corrupt distance maps with 30 errors. We repeated this simulation 10 times. (c,d) The distribution function of change in the loss term in response to injecting 30 errors. In (c), $C$ is evaluated using the filtration by thresholding distance maps, whereas in (d) we use our filtration. The probability of decreasing the existing loss term by injecting additional errors is around $0.4$, whereas for our loss term it drops to $0.2$. We conclude that our loss term is more monotonic with respect to the error number.

In its simplest form, $g$ is a linear function of pixel coordinates, and the region highlighted for any $s$ extends along a line perpendicular to the height axis, as shown in Fig. 3(c). But other forms of $g$ are also possible. We tested

- linear functions $g(\mathbf{p}) = \mathbf{w}^\intercal \mathbf{p}$, where $\mathbf{w}$ is a two-vector hyper-parameter encoding the orientation of the height axis and the slope of the height function;
- a scaled distance to a point $\mathbf{q}$ in the image, $g(\mathbf{p}) = a\|\mathbf{p}-\mathbf{q}\|_2$, where $\mathbf{q}$ and $a$ are hyper-parameters;
- the square of the height function $g(\mathbf{p}) = \mathbf{p}^\intercal \mathbf{W} \mathbf{p}$, where $\mathbf{W} = \mathbf{w}^\intercal \mathbf{w}$, and $\mathbf{w}$ is the hyper parameter encoding the slope of the function and the orientation of the height axis;

The function $g$ introduces partial information of location of topological features into the persistence diagram. This is illustrated by Fig. 3 where different values of the scale parameter make homology classes appear in different parts of the image. But, because the scale parameter must be a scalar, it can only pinpoint location of topological features in 2D or 3D images along one direction. This could be addressed by evaluating the loss function many times for many different orientations of the height axis, or more generally, for many different hyper-parameters of $g$. This approach is legitimized by the theoretical result by (Betthauser, 2018), who proved that four well chosen filtration directions suffice to completely represent a binary image. The problem of combining a number of different filtration functions is known in topological literature as multipersistence (Carlsson & Zomorodian, 2009). But current multipersistence techniques are not easily plugged into a deep learning framework due to the lack of results on their differentiability. Moreover, filtering the data along multiple directions would considerably slow down training. Instead, we randomly draw the hyper-parameters of the height function at each training iteration. We show in the supplementary material that, in practice, the simple linear function performs best.

### 3.4 VALIDATION ON SYNTHETIC DATA

We motivated our filtration technique by the fact that it introduces partial localization of topological features into the persistence diagrams and better spreads the diagrams across the plane. We validated it on synthetic data to show that it correlates better with the number of errors injected into a distance map than the baseline loss based on thresholding distance maps. To that end, we took two crops of ground truth road graphs of the *RTracer* dataset (Bastani et al., 2018) and generated faulty synthetic distance maps by injecting one error at a time, randomly selected between a road disconnection and a false interconnection with equal probability. We then evaluated the topological discrepancy $C$ equation 1 using either filtration by thresholding distance maps, or our combined filtration. We

plotted the distribution function of the change in $C$ resulting from error injection in Fig. 4. When using the standard approach, injecting new errors is likely to decrease this loss term. Our approach results in a twofold reduction of this probability.

## 4 EXPERIMENTS

We now describe the dataset we have tested our approach on, the baselines to which we compare our results, and the metrics we used to assess the topological correctness of the segmentations. We then demonstrate that our new loss improves the topological correctness of segmentation masks. We provide additional qualitative results and an ablation study in the supplementary material.

### 4.1 DATASETS

We experimented on three datasets.

- *RTracer*. A recently published dataset of high-resolution satellite images covering urban areas of forty cities in six different countries (Bastani et al., 2018). The ground truth was obtained from OpenStreetMap. Like (Bastani et al., 2018; Li et al., 2019; Yang et al., 2019; Mosińska et al., 2020), we used twenty five cities as the training set and the remaining fifteen as the test set.

- *Massachusetts*. The Massachusetts dataset (Mnih, 2013) features both urban and rural neighborhoods, with many different kinds of roads ranging from small paths to highways. For a fair comparison to (Hu et al., 2019), we split the data into three equal folds and performed a three-way cross validation.

- *Neurons*. The dataset is a part of a proprietary 3D, 2-photon microscopy scan of a whole mouse brain. It contains 14 stacks of size $250 \times 250 \times 200$ voxels and a spatial resolution of $1.0 \times 0.3 \times 0.3$ $\mu m$. We used ten stacks for training and the remaining four for testing.

- *Brain*. The dataset contains two 3D images of neurons in a mouse brain. The axons and dendrites have been outlined manually while viewing the sample under a microscope and the image has been captured later. The sample deformed in the meantime, resulting in a misalignment between the annotation and the image. We use twelve stacks of size $150 \times 200 \times 200$ voxels and a spatial resolution of $1$ $\mu m$ for training and ten of them for testing.

### 4.2 METHODS TESTED

To test the impact of our proposed filtration functions, we used the standard U-Net architecture (Ronneberger et al., 2015), with four blocks, each with two sequences of convolution-ReLU-batch normalization. Max-pooling in $2 \times 2$ windows followed each of the blocks. The initial feature size was set to 32 and grew to 512 in the smallest feature map in the network. We augmented the training data with vertical and horizontal flips and random rotations and used the ADAM algorithm (Kingma & Ba, 2015) with the learning rate set to $1e - 4$. We then used different version of the $L_{tot}$ of Eq. 1 we minimized to train the network. We tested the following as baselines:

- *UNet-CE*. $L$ is the Cross Entropy loss for pixel classification and there is no topological discrepancy loss, that is, $\alpha = 0.0$.

- *UNet-MSE*. $L$ is the mean squared error of the truncated distance to the closest foreground pixel, with no topological discrepancy loss.

- *Homo-Pre*. $L$ is the cross Entropy loss and we compute $C$ by thresholding pixel classification maps., as in (Hu et al., 2019; Clough et al., 2019; 2020).

- *Homo-Reg*. $L$ is the mean squared error and we compute $C$ by thresholding the truncated distance maps, as in Wang et al. (2020).

- *Homo-Ours*. $L$ is the mean squared error and we compute $C$ using our proposed filtration function.

In the last three cases, we set $\alpha$ to 0.01 for all our experiments. Like Hu et al. (2019), we compute the loss in windows sized $64 \times 64$ pixels, and limit the method to homology classes order 1, that is, loops. This has two advantages. First, by convention, loops are created by the borders of the window, making disconnections in dead-ending roads or neurites detected as broken loops. Second, detection of homology classes is computationally expensive, and the time grows cubically with the number of pixels. In our current setup, computing the loss for a single window takes 0.5 seconds. Similarly to (Hu et al., 2019), we did not observe any performance gain due to using homology classes of order 0—connected components—in addition to loops.

For completeness, we also compared our approach to recent techniques *not* relying on persistent homologies *Segmentation* (Bastani et al., 2018), *RoadTracer* (Bastani et al., 2018), *Seg-Path* (Mosińska et al., 2020), *RCNNU-Net* (Yang et al., 2019), *DeepRoad* (Máttyus et al., 2017), *PolyMapper* (Li et al., 2019), *DMT* (Hu et al., 2021), and *ConnLoss* (Oner et al., 2021). *Segmentation*, *RoadTracer*, *RCNNU-Net*, and *PolyMapper* do no explicitly enforce topology constraints, while the others do and are discussed in the related work section. The outputs of these methods were shared by the authors directly with us or on the Internet, and we computed all the performance metrics.

## 4.3 PERFORMANCE METRICS

Comparing connectivity of segmentation masks is difficult, because the reconstructions rarely overlap with the ground truth, and often deviate from it significantly. There seems to be no consensus concerning the best evaluation technique; we found five connectivity-oriented metrics in concurrently published recent work. To provide an exhaustive evaluation, we used all of them.

- *APLS* for Average Path Length Similarity. It is defined as an aggregation of relative length difference of shortest paths between pairs of corresponding points in the ground truth and predicted maps Etten et al. (2018).

- *TLTS*. It is a statistics of lengths of shortest paths between corresponding pairs of end points randomly selected in the predicted and ground-truth networks Wegner et al. (2013). We report the fraction of paths for which the relative length difference is within 5%.

- *JCT*. It is a junction score that considers the number of roads intersecting at each junction Bastani et al. (2018). It consists of road recall, averaged over the intersections of the ground-truth and road precision, averaged over the intersections of the prediction. We report the corresponding F1 score.

- *Betti*. The Betti error, as defined by (Hu et al., 2019), is an average absolute difference between the number of topological structures seen in the ground truth and predicted delineations. We take random patches sized $64 \times 64$ from predictions, compute the number of 1-homology classes (loops) and compare the numbers computed for the prediction and the ground truth. We average this difference over 10 trials. In practice, to compute the error we use the code made publicly available by the authors.

- *CCQ* We complement the connectivity-oriented with the most popular metric that measures spatial co-occurrence of annotated and predicted road pixels, rather than connectivity. The Correctness, Completeness and Quality are equivalent to precision, recall and intersection-over-union, where the definition of a true positive has been relaxed from spatial coincidence of prediction and annotation to co-occurrence within a distance of 5 pixels Wiedemann et al. (1998). We report the Quality as our single-number metric.

Table 1: On the *Massachusetts* dataset, our loss function outperforms all PH-based loss functions. The results for our method are means and standard deviations over three independent training runs.

| Method | Connectivity-oriented | | | | pixel-based |
| | *APLS* | *TLTS* | *JCT* | *Betti* | *CCQ* |
|---|---|---|---|---|---|
| *UNet-CE* | $60.9 \pm 3.9$ | $41.6 \pm 4.1$ | $72.0 \pm 2.7$ | $3.12 \pm 0.6$ | $66.9 \pm 2.6$ |
| *UNet-MSE* | $61.3 \pm 3.7$ | $41.9 \pm 4.2$ | $71.9 \pm 2.9$ | $3.09 \pm 0.7$ | $67.3 \pm 2.3$ |
| *DMT* | $64.7 \pm 2.9$ | $45.8 \pm 2.8$ | $80.6 \pm 2.4$ | $\mathbf{0.99} \pm 0.4$ | $74.9 \pm 1.9$ |
| *ConnLoss* | $\mathbf{73.4} \pm 3.6$ | $\mathbf{53.2} \pm 4.4$ | $\mathbf{81.4} \pm 1.9$ | $1.29 \pm 0.5$ | $\mathbf{75.8} \pm 2.2$ |
| *Homo-Pre* | $62.5 \pm 1.9$ | $42.1 \pm 1.9$ | $74.2 \pm 1.7$ | $1.28 \pm 0.3$ | $69.3 \pm 1.9$ |
| *Homo-Reg* | $65.0 \pm 2.2$ | $45.6 \pm 1.8$ | $76.9 \pm 1.9$ | $1.09 \pm 0.2$ | $71.8 \pm 2.1$ |
| *Homo-Ours* | $\mathbf{68.7} \pm 1.2$ | $\mathbf{50.6} \pm 2.3$ | $\mathbf{79.2} \pm 2.6$ | $\mathbf{0.90} \pm 0.3$ | $\mathbf{74.9} \pm 1.8$ |

## 4.4 COMPARATIVE RESULTS

As shown in Tabs 1 and 2, on the *Massachusetts* and *RTracer* data sets, our method outperforms the other methods based on Persistent Homology, which demonstrates that our approach to filtering is truly effective. It also outperforms the other 2D tracing algorithms targeted at handling aerial images, *RoadTracer*, *Seg-Path*, *DeepRoad*, and *PolyMapper*, at the exception of *ConnLoss* that does marginally better. This is presumably because *ConnLoss* explicitly penalizes each disconnection of the prediction, whereas a persistence diagram is a lossy topological descriptor that may fail to

Table 2: Our loss function outperforms all PH-based loss functions on the *RTracer* dataset. Means and standard deviations over cities from the test set are reported.

| Method | Connectivity-oriented | | | | pixel-based |
|---|---|---|---|---|---|
| | *APLS* | *TLTS* | *JCT* | *Betti* | *CCQ* |
| *UNet-CE* | $63.4 \pm 1.6$ | $37.5 \pm 1.9$ | $78.0 \pm 1.0$ | $3.08 \pm 0.6$ | $59.7 \pm 2.2$ |
| *UNet-MSE* | $66.3 \pm 1.9$ | $40.0 \pm 2.0$ | $77.5 \pm 1.3$ | $2.99 \pm 0.5$ | $59.5 \pm 1.9$ |
| *Segmentation* | $62.5 \pm 1.5$ | $33.0 \pm 1.6$ | $78.2 \pm 1.5$ | $3.04 \pm 0.6$ | $54.4 \pm 1.0$ |
| *RoadTracer* | $59.1 \pm 0.8$ | $40.6 \pm 1.5$ | $81.2 \pm 1.6$ | $2.85 \pm 0.7$ | $47.8 \pm 1.6$ |
| *Seg-Path* | $68.1 \pm 1.4$ | $46.5 \pm 1.7$ | $75.4 \pm 1.3$ | $2.31 \pm 0.4$ | $54.0 \pm 1.4$ |
| *RCNNU-Net* | $48.2 \pm 1.6$ | $18.4 \pm 1.9$ | $75.9 \pm 1.4$ | $3.25 \pm 0.7$ | $62.8 \pm 1.5$ |
| *DeepRoad* | $24.6 \pm 2.2$ | $6.4 \pm 0.9$ | $51.4 \pm 1.5$ | $4.95 \pm 1.0$ | $43.6 \pm 2.0$ |
| *PolyMapper* | $61.3 \pm 2.3$ | $31.5 \pm 1.9$ | $80.0 \pm 1.2$ | $2.90 \pm 0.4$ | $35.7 \pm 1.4$ |
| *ConnLoss* | $\mathbf{75.4} \pm 1.6$ | $\mathbf{49.6} \pm 1.4$ | $\mathbf{82.6} \pm 0.6$ | $\mathbf{1.30} \pm 0.4$ | $\mathbf{68.4} \pm 0.9$ |
| *Homo-Pre* | $67.3 \pm 1.7$ | $42.3 \pm 1.1$ | $78.7 \pm 0.9$ | $1.32 \pm 0.3$ | $61.9 \pm 1.9$ |
| *Homo-Reg* | $69.9 \pm 1.6$ | $45.1 \pm 1.4$ | $79.6 \pm 1.3$ | $1.07 \pm 0.3$ | $63.2 \pm 1.6$ |
| *Homo-Ours* | $\mathbf{73.8} \pm 1.8$ | $\mathbf{47.8} \pm 0.9$ | $\mathbf{81.3} \pm 1.6$ | $\mathbf{0.89} \pm 0.2$ | $\mathbf{66.3} \pm 1.7$ |

Table 3: Comparative results on the *Neurons* dataset. Our loss outperforms all the baselines. The results for our method are means and standard deviations over three independent training runs.

| Method | Connectivity-oriented | | | pixel-based |
|---|---|---|---|---|
| | *APLS* | *TLTS* | *Betti* | *CCQ* |
| *UNet-CE* | $79.9 \pm 1.5$ | $80.8 \pm 2.2$ | $2.33 \pm 0.6$ | $90.6 \pm 2.0$ |
| *UNet-MSE* | $80.2 \pm 1.6$ | $80.9 \pm 2.0$ | $2.31 \pm 0.7$ | $90.4 \pm 1.9$ |
| *Homo-Pre* | $83.5 \pm 1.0$ | $82.1 \pm 1.7$ | $1.06 \pm 0.2$ | $91.2 \pm 1.8$ |
| *Homo-Reg* | $85.4 \pm 1.2$ | $83.4 \pm 1.5$ | $0.91 \pm 0.2$ | $92.5 \pm 1.6$ |
| *Homo-Ours* | $\mathbf{86.9} \pm 1.1$ | $\mathbf{85.2} \pm 1.9$ | $\mathbf{0.80} \pm 0.2$ | $\mathbf{93.3} \pm 1.9$ |

Table 4: Comparative results on the *Brain* dataset. Our loss outperforms all PH-based losses. The results for our method are means and standard deviations over three independent training runs.

| Method | Connectivity-oriented | | | pixel-based |
|---|---|---|---|---|
| | *APLS* | *TLTS* | *Betti* | *CCQ* |
| *UNet-CE* | $65.8 \pm 1.8$ | $63.6 \pm 1.3$ | $2.89 \pm 0.4$ | $70.4 \pm 1.9$ |
| *UNet-MSE* | $66.0 \pm 1.6$ | $63.9 \pm 1.4$ | $2.92 \pm 0.5$ | $70.6 \pm 1.8$ |
| *Homo-Pre* | $67.6 \pm 1.5$ | $65.3 \pm 1.0$ | $1.39 \pm 0.2$ | $71.5 \pm 1.4$ |
| *Homo-Reg* | $70.5 \pm 1.5$ | $68.8 \pm 0.9$ | $1.22 \pm 0.3$ | $72.6 \pm 1.3$ |
| *Homo-Ours* | $\mathbf{73.4} \pm 1.4$ | $\mathbf{70.1} \pm 1.1$ | $\mathbf{1.06} \pm 0.2$ | $73.2 \pm 1.2$ |

penalize some errors. However, *ConnLoss* does not naturally extend to 3D data, whereas our method does. On the 3D *Neurons* data set, it outperforms the competing algorithms, as evidenced by the results shown in Tab. 3. We provide qualitative results in the supplementary material.

## 5 CONCLUSION

We proposed an improved approach to using Persistent Homology to train deep networks to delineate curvilinear structures. It outperforms current such approaches by introducing an element of location in the filtration process. Unlike other powerful approaches (Oner et al., 2021) to enforcing topological constraints on the output of deep networks, it generalizes naturally to 3D, which opens the door to future research in a space that is critical for biomedical applications.

To further increase performance, we will address the fact that our proposed loss function has sparse gradients that only depend on values at pixels that are critical for emergence and disappearance of topological features. This limits robustness and our future work will focus on developing more sophisticated topological descriptors with more smooth gradients.

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

# A   SUPPLEMENTARY MATERIAL

## A.1   QUALITATIVE RESULTS

In this section, we provide qualitative results on our three test datasets. For each method, we display the thresholded predictions with their skeletons overlaid in red. In the case of the 3D dataset, the images we show are maximum intensity projections.

## A.2   ABLATION STUDY

To investigate the impact of hyper-parameter choices on performance, we ran three ablation studies.

**Weighting the PH Loss**   We have varied the coefficient $\alpha$ in equation 1, while keeping the other parameters fixed. We report the results in Tab. 5. The best results are achieved for $\alpha = 0.01$, and the performance decreases when $\alpha$ is set ten times higher or lower. This suggests that the standard Mean Square Loss is still important for overall performance, which is not a surprise as the gradient of our persistent-homology-based loss is sparse and concentrated at pixels critical for topological correctness.

**Window size**   We changed the size of the window in which the persistent homology is computed. We report the results in Tab. 6. Our method performs best when using large windows that contain

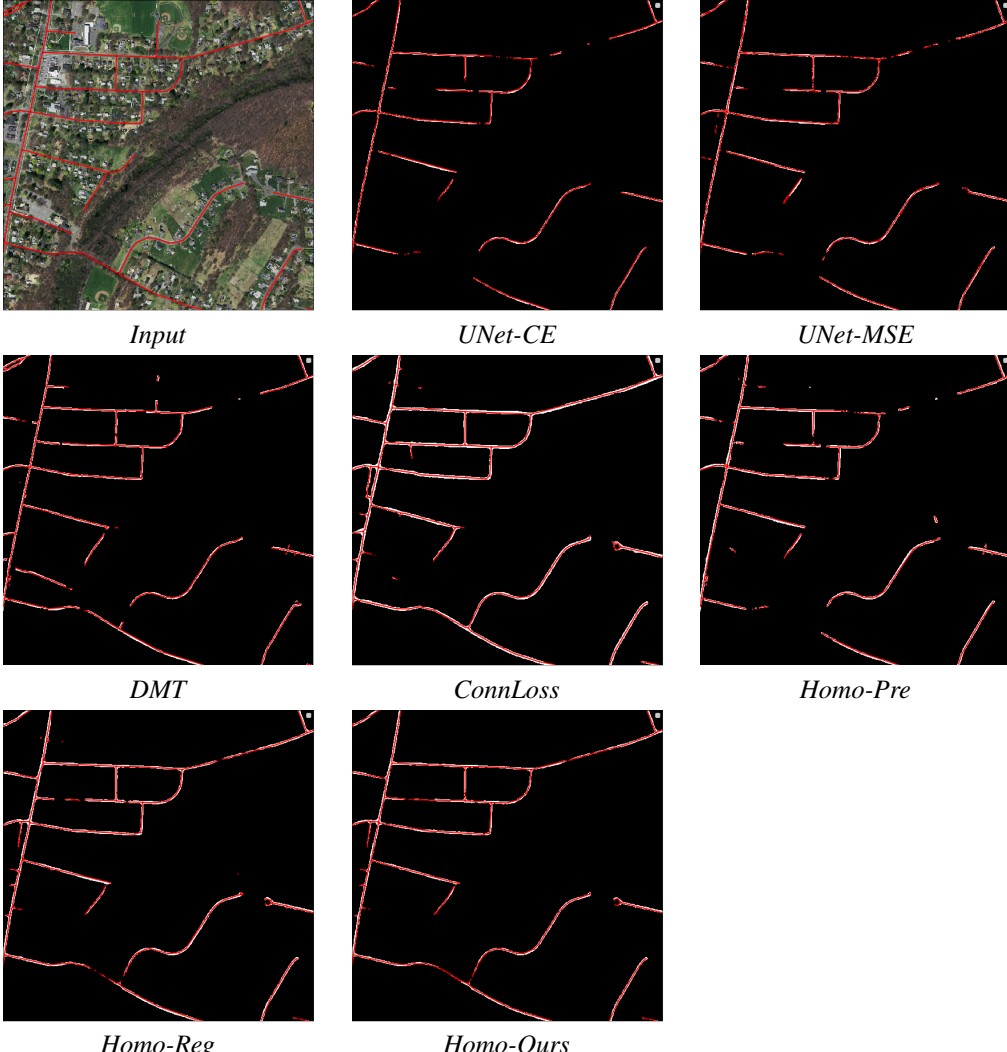

Figure 5: Comparative results on the *Massachusetts* dataset.

significant portions of the structures of interest. We could not try even larger ones because it would have increased the time needed to detect the homologies and slowed down the training too much.

**Height Functions**    We also evaluated the effect on performance of using different forms of function $g$ in equation 2, that ties homology birth and death times to image coordinates, distributing the points in persistence diagram. We present the results in Tab. 7. The distance to a random image point, or the

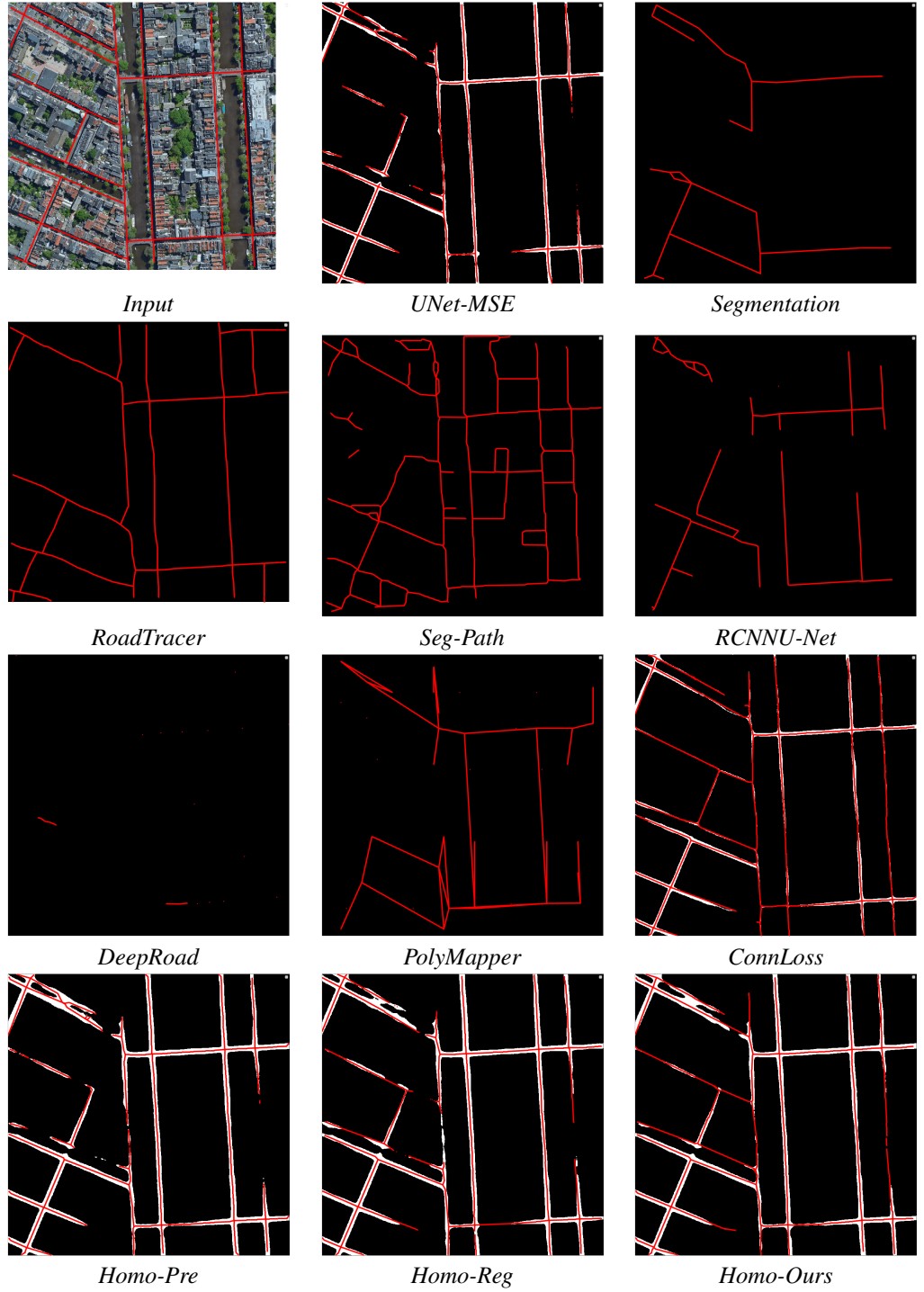

Figure 6: Comparative results on the *RTracer* dataset.

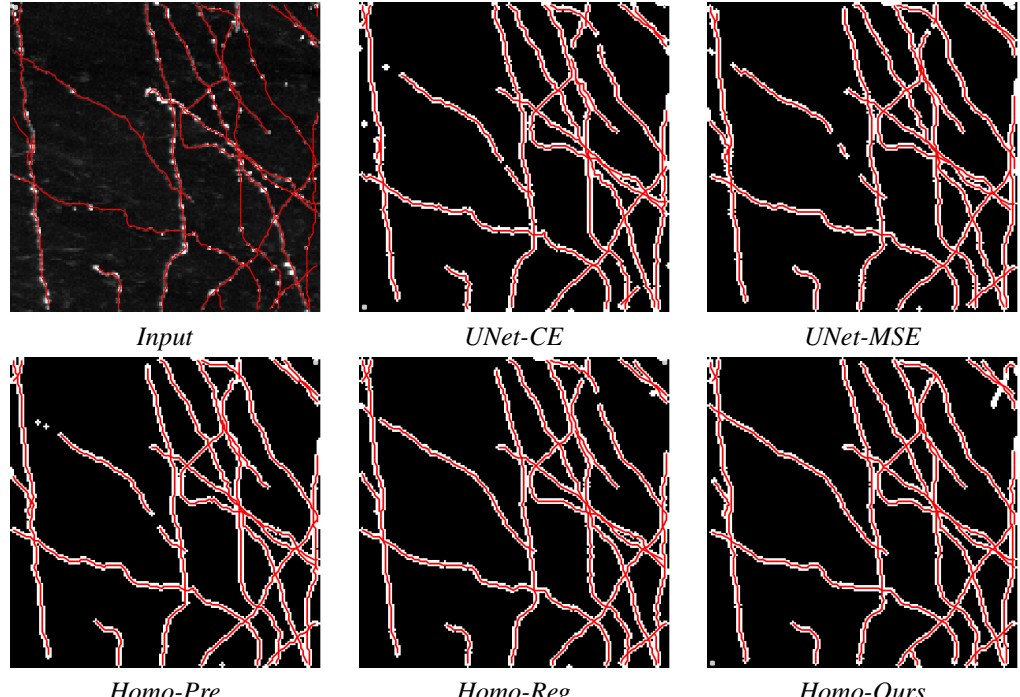

| *Input* | *UNet-CE* | *UNet-MSE* |
| *Homo-Pre* | *Homo-Reg* | *Homo-Ours* |

Figure 7: Comparative results on the 3D *Neurons* dataset.

Table 5: Impact of changing the learning coefficient of localized PH loss on the *Massachusetts* dataset. The window size is fixed to 64x64.

| $\alpha$ | Connectivity-oriented | | | | pixel-based |
| | APLS | TLTS | JCT | Betti | CCQ |
|---|---|---|---|---|---|
| 1e-3 | 64.9 | 46.0 | 77.1 | 1.21 | 72.3 |
| 1e-2 | **68.7** | **50.6** | **79.2** | **0.90** | **74.9** |
| 1e-1 | 67.1 | 48.9 | 77.8 | 0.94 | 74.6 |
| 1e-0 | 64.8 | 45.8 | 76.2 | 1.10 | 72.0 |

Table 6: Impact of changing the window size when computing our localized loss on the *Massachusetts* dataset. The learning coefficient is fixed to 1e-2.

| Window Size | Connectivity-oriented | | | | pixel-based |
| | APLS | TLTS | JCT | Betti | CCQ |
|---|---|---|---|---|---|
| 8x8 | 62.1 | 41.9 | 73.0 | 2.84 | 67.2 |
| 16x16 | 62.7 | 42.4 | 74.5 | 2.09 | 68.8 |
| 32x32 | 65.4 | 45.7 | 77.1 | 1.17 | 72.5 |
| 64x64 | **68.7** | **50.6** | **79.2** | **0.90** | **74.9** |

use of a quadratic instead of linear function of image coordinates do not result in higher performance than the plain linear function.

Table 7: Performances of different height functions used for localized PH loss on the *Massachusetts* dataset. The learning coefficient is fixed to 1e-2 and window size to 64x64

| Height Function | Connectivity-oriented | | | | pixel-based |
|---|---|---|---|---|---|
| | *APLS* | *TLTS* | *JCT* | *Betti* | *CCQ* |
| Distance to a point | 67.8 | 49.4 | 77.9 | 1.01 | 73.6 |
| Random Linear | **68.7** | **50.6** | **79.2** | **0.90** | **74.9** |
| Fixed Linear | 67.5 | 48.7 | 76.5 | 1.15 | 73.0 |
| Square | 64.2 | 45.1 | 76.3 | 1.32 | 70.3 |

