# OpenReview forum: "Localized Persistent Homologies for more Effective Deep Learning"
_ICLR.cc/2022/Conference — ICLR 2022 Submitted_

### Official Review · Reviewer_Emfj · 2021-10-29

**Correctness:** 3
**Technical Novelty And Significance:** 2
**Empirical Novelty And Significance:** 3
**Recommendation:** 6
**Confidence:** 4

**Main Review:**

# Strengths and Weaknesses

## Strengths
- It is nice to see a well-motivated application of TDA in the context of image processing.
- Overall, the flow of the work is clear and the paper is pleasant to read.
- The experimental results reported seem solid, extensive, and may be a good contribution to the field of image segmentation in particular.

## Weaknesses
- The paper contains some caveats / inaccuracies related to TDA (see the Minor Comments section for details). In particular, the desire to filter an object with respect to two (or more) parameters $Y, g$ (here, distance + height) is present for a long-time in the TDA community, known as *multi-persistence* (which is way harder than single-parameter persistence and, roughly, untractable in practice for now). Simplifying this by instead looking at one (or many!) linear combination of the two maps $Y, g$ is already used in TDA, see for instance the work *Multiparameter Persistence Images for Topological Machine Learning* by Carrière and Blumberg (2020); if my understanding is correct, filtering with respect to $Y + g$ is very similar (if not the same) to picking one direction in the so-called *fibered barcode*. Even if this does not take away all the potential impact of the paper (and it is quite nice to think about **this** choice of $Y,g$ and that it seemingly works well in applications), I think this must be discussed and the position of the paper with respect to the TDA literature should be revised a bit.

# Minor comments
- "Persistent homologies" (abstract), I think it is not standard to use plural here. "Persistent homology" would fit better in my opinion.
- (Ref) When introducing Persistent Homology (p1, third paragraph), I think it may be appropriate to cite (in addition) a founding paper of TDA or a reference book, such as the one(s) of G.Carlsonn & Zomorodian (e.g. *Computing Persistent Homology* (2005)), Edelsbrunner and Harer (2008), Chazal and Michel (2017), etc.
- (Ref) About the possibility to  differentiate the map $X \mapsto \mathrm{Dgm}(X)$ (should $X$ be an image or something else), I think it may be worth citing the work *A topology layer for machine learning* by Bruel-Gabrielsson et al. (possibly along with other work, like *A framework for differential calculus on persistence barcodes* by Leygonie et al., and *Optimizing persistent homology based functions* by Carrière et al.).
- (Typo) Caption of Figure 1, Something is wrong, a description of *(d)* is missing.
- (Ref) Saying that "Application of computer vision [of persistent homology] were scarce" may be slightly exaggerated, see for instance the short survey *Topological Data Analysis in Computer Vision* by Bernstein et al. and the references therein.
- (Typo) "closed 3D surfaces", I guess "surfaces embedded in 3D" may be more accurate.
- (Ref) Citing Giotto (Tauzin et al.) for the "matrix reduction algorithm" to compute PH is not very accurate as well. This algorithm was introduced much earlier (see references at the beginning of https://en.wikipedia.org/wiki/Topological_data_analysis#Computation ). Of course, it is worth mentioning the numerical computations were done using Giotto though.
- (Important) p4, "The cost of leaving a homology unmatched" is **not** constant (at least in standard definition of the distances between PDs). It is the distance between the point and its orthogonal projection on the diagonal, as drawn for instance on Figure 3. If you used a constant cost instead, this must be discussed and motivated.
- Calling "a homology $g \in H_{X_1}$..." a point in the persistence diagram is not accurate. Calling them "A persistence interval" as done few lines above is much more standard in my opinion.
- Figure 4 and Section 3.4 are not fully convincing : though *(c)* is indeed non-monotonic, the trend seems similar (and perhaps more trials than just 5 would have smooth the curve. Plotting the standard deviation would be useful as well). I would not say that "very noisy images can be seen as being very similar to error-free ones."

**Summary Of The Paper:**

Relying on recent progress regarding the (automatic) differentiation of « Persistent Homology » (a tool of Topological Data Analysis (TDA) to extract topological descriptors on top of complex data), this paper proposes a new filtration (essentially, a sum of the distance filtration and the height filtration) to favor proper topological reconstruction in the context of image processing (segmentation in particular).
The authors experimentally showcase their approach on different (2D and 3D) datasets and compare it with various competitors (from and outside TDA-based techniques) and obtain very competitive results.

**Summary Of The Review:**

The paper is well-written and showcases an interesting application of TDA in the context of image processing supported by various numerical experiments. However, it contains some inaccuracies regarding TDA concepts/literature that must be corrected/discussed.

---

> ### Author Response · Authors · 2021-11-19
> **Response to the comments of Reviewer Emfj**
>
> Thank you for the review. The changes to the text of the paper are marked in blue.
>
> Q1: The paper contains some inaccuracies related to TDA (see the Minor Comments section for details).
>
> A1: We have corrected the inaccuracies. Thank you for pointing them out.
>
> Q2: (...) the desire to filter an object with respect to two (or more) parameters Y,g (here, distance + height) is present for a long-time in the TDA community, known as multi-persistence (which is way harder than single-parameter persistence and, roughly, intractable in practice for now). Simplifying this by instead looking at one (or many!) linear combination of the two maps Y,g is already used in TDA, see for instance the work Multiparameter Persistence Images for Topological Machine Learning by Carrière and Blumberg (2020); if my understanding is correct, filtering with respect to Y+g is very similar (if not the same) to picking one direction in the so-called fibered barcode. Even if this does not take away all the potential impact of the paper (and it is quite nice to think about this choice of Y,g and that it seemingly works well in applications), I think this must be discussed and the position of the paper with respect to the TDA literature should be revised a bit.
>
> A2: We thank the reviewer for pointing out the paper of Carriere and Blumberg, we now reference it in the manuscript.
> Indeed, when applied to binary images, the filtration $Y+g$ is a line in the fibered barcode. Because several directions are considered, the problem here does not fall into the 2-parameter persistence framework, and the tools developed in this paper (multiparameter persistence images) cannot be directly applied. Using multiparameter persistence combined with deep learning seems, for now, to be too complicated and we choose to simplify the problem by combining two filtrations. We now include this discussion in the manuscript.
>
> Q3: "Persistent homologies" (abstract), I think it is not standard to use plural here. "Persistent homology" would fit better in my opinion.
>
> A3: We changed the title and the abstract.
>
> Q4: (Ref) When introducing Persistent Homology (p1, third paragraph), I think it may be appropriate to cite (in addition) a founding paper of TDA or a reference book, such as the one(s) of G.Carlsonn & Zomorodian (e.g. Computing Persistent Homology (2005)), Edelsbrunner and Harer (2008), Chazal and Michel (2017), etc.
>
> A4: We added references to surveys in persistent homology and persistent homology for images. We also made the introduction to persistent homology more precise.
>
> Q5: (Ref) About the possibility to differentiate the map X->Dgm(X) (should X be an image or something else), I think it may be worth citing the work A topology layer for machine learning by Bruel-Gabrielsson et al. (possibly along with other work, like A framework for differential calculus on persistence barcodes by Leygonie et al., and Optimizing persistent homology based functions by Carrière et al.).
>
> A5: We added references to these papers and we now explain the timeline of differentiating persistent homology better.
>
> Q6: (Ref) Saying that "Application of computer vision [of persistent homology] were scarce" may be slightly exaggerated, see for instance the short survey Topological Data Analysis in Computer Vision by Bernstein et al. and the references therein.
>
> A6: Right. We changed this fragment. We now explain that, before the work of Hu et al. and Clough et al., PH was not used as a loss function to train deep networks.
>
> Q7: (Typo) "closed 3D surfaces", I guess "surfaces embedded in 3D" may be more accurate.
>
> A7: We changed it. Thank you.
>
> Q8: (Ref) Citing Giotto (Tauzin et al.) for the "matrix reduction algorithm" to compute PH is not very accurate as well. This algorithm was introduced earlier (...).
>
> A8: We now cite Edelsbrunner and Harer, and also Gudhi for the computations (since Giotto uses the cubical complex computations of Gudhi). We added a sentence to cite cubical complexes for images using [https://arxiv.org/pdf/2005.04597.pdf].
>
> Q9: Cost of homology unmatched is not constant.
>
> A9: Thank you, we corrected this error.
>
> Q10: Homology is not a point in persistence diagram.
>
> A10: Thank you for pointing out this error.
>
> Q11: Figure 4 and Section 3.4 are not fully convincing : though (c) is indeed non-monotonic, the trend seems similar (and perhaps more trials than just 5 would have smooth the curve. Plotting the standard deviation would be useful as well). I would not say that "very noisy images can be seen as being very similar to error-free ones."
>
> A11: We have modified the figure and we present the likelihood that injecting errors decreases the loss term. This method inherently represents standard deviation and we hope the Reviewer will find it more convincing.

---

> > ### Comment · Reviewer_Emfj · 2021-11-19
> > **Thanks**
> >
> > Thank you for taking time to answer my comments.

---

### Official Review · Reviewer_jz5f · 2021-10-30

**Correctness:** 4
**Technical Novelty And Significance:** 3
**Empirical Novelty And Significance:** 3
**Recommendation:** 8
**Confidence:** 4

**Main Review:**

The paper addresses an important problem of training network producing segmentation with desired topology. Traditionally, segmentation is done via pixel-wise losses, which are not able to properly capture topology, resulting in significant topological mistakes such as lack of connectivity or erroneous connections. Designing topologically-motivated losses, like in this paper, is an important problem for detecting objects like roads, vasculature, etc.

The paper builds upon an established framework of persistent homologies (PH) and extends it by adding positional embedding.

The results look convincing and the proposed modification seems simple and effective.

The topological loss based on PH is quite complex in the implementation. The paper would benefit from a more detailed review of the implementation including differentiation.

The author should discuss the relation and compare to other PH-based localization techniques, for example Hu et al (2019). In particular Hu et al used patch-based topology, which provides a degree of localization. How does this compare to the proposed method? DO the authors explicitly compare to the patch based approach?



**Summary Of The Paper:**

The proposes a localized version of topological loss. Current methods perform matching in the space (i.e. persistence diagrams) that does not account for the location of topological features. The method proposes to add to the likelihood maps some sort of positional embedding, i.e. a function of image pixel coordinates chosen at random during training. The experiments demonstrate improved performance.

**Summary Of The Review:**

The paper is addresses an important problem of topology-aware segmentation learning. The proposed framework extends previously existing PH diagram techniques by adding positional emebedding, this add locality to the framework. The experimental evaluation shows improvement upon global methods. Some lack of comparison with other localized methods.

---

> ### Author Response · Authors · 2021-11-19
> **Response to the comments of Reviewer jz5f**
>
> Thank you for reviewing the paper. We marked the changes to the manuscript in blue.
>
> Q1: The paper would benefit from a more detailed review of the implementation including differentiation.
>
> A1: For differentiating the persistence diagram, we use code made publicly available by Hu et al. (2019) The details of this differentiation are described in their paper. The main idea consists in assigning the gradients of $\pm 1$ to the pixels responsible for emergence and disappearance of topological features during filtration, for example, the value of a pixel that closes a loop is the left boundary of its persistence interval, and the value of the pixel that fills the loop is the right boundary of this interval.
>
> Q2: Hu et al. used patch-based topology, which provides a degree of localization. How does this compare to the proposed method?
>
> A2: This is an excellent question. Indeed, Hu et al. use a patch-based approach, to which we directly compare. Our method also exploits the same patch-based approach. Evaluating PH in patches is currently a necessity due to: 1) the time needed to compute the persistence diagrams 2) the fact that the matching of persistence diagrams is much more likely to be incorrect when many topological features are present in the diagrams. The patch-based approach does provide a degree of localization, but much coarser than the localization implemented in our loss function. In fact, the localization brought in by our filtration function helps to disambiguate between the persistence intervals obtained for a single patch.

---

> > ### Comment · Reviewer_jz5f · 2021-11-29
> > **Thank you for addressing my questions**
> >
> > I maintain my rating for the paper

---

### Official Review · Reviewer_mHrP · 2021-10-31

**Correctness:** 3
**Technical Novelty And Significance:** 2
**Empirical Novelty And Significance:** 2
**Recommendation:** 3
**Confidence:** 5

**Main Review:**

While I find the topic of this paper highly fascinating, at present,
this paper is not yet ready for publication. There are three main issues
I have with the current write-up:

1. The work is marred by severe deficiencies in clarity/correctness.
   I will provide more examples later but the text is using several
   terms incorrectly and does not explain underlying concepts correctly.

2. Missing delineation to existing work: while the paper discusses other
   work that aims to combine image tasks with topological prior
   information, it does not provide a sufficiently detailed delineation
   to such work. This can also be seen in the related work section,
   which misses the paper [*A Topological Loss Function for Deep-Learning
   based Image Segmentation using Persistent
   Homology*](https://arxiv.org/abs/1910.01877) by Clough et al.
   (linking to the preprint here; the full paper appears in IEEE TPAMI).
   While another paper of Clough et al. is discussed, the aforementioned
   work should also be at briefly discussed and, ideally, be used as
   a comparison partner.

3. The experimental setup is preliminary and missing important details:
   means and standard deviations are only provided for the proposed
   method, whereas the numbers for other methods are presumably cited
   from an additional publication. This does not enable a fair
   comparison, given the fact that it is unclear whether the experiments
   described in other papers followed the same training regimen.
   Providing standard deviations is a critical facet that needs to be
   changed prior to publication.

   In addition, I find the selection of comparison metrics somewhat
   questionable; in other publications, metrics such as accuracy, DICE
   scores, ARI, etc. are being employed, whereas this paper reports
   metrics that appear to be relatively non-standard to me. That being
   said, I acknowledge that I am not familiar with all segmentation
   literature; I am, however, very familiar with the segmentation
   literature based on topological concepts, and these papers
   consistently report different scores. I would therefore suggest, also
   in the interest of making the paper more accessible, to include
   additional scores and expand on existing ones (for instance, it would
   be helpful to know what a 'good' score looks like, and whether larger
   numbers are better or worse. All of this information can be gleaned
   from the text at present but it could be more prominent).

   Finally, I am confused by certain reported numbers and algorithms in
   the experimental section:

   - It seems that `ConnLoss` is performing better in all metrics.
     I understand that it does not extend to 3D data, but if this is to
     be the deciding property of the proposed algorithm, additional
     experiments on 3D data sets are required. Performance on the
     proprietary data set is hard to assess because on comparison
     partner papers exist. While it is perfectly acceptable to use such
     data sets, in the interest of a fair comparison, I would suggest to
     also employ additional 3D data sets such as `ISBI13`, `CREMI`, or
     `3Dircadb`, all of which are discussed in [the paper by Hu et
     al.](https://arxiv.org/pdf/2103.09992.pdf).

   - The 'Massachusetts' data set corresponds to the `ROAD` data set from
    the aforementioned paper by Hu et al. However, Hu et al. report the
    same Betti error as the one shown in this paper, albeit with
    a standard deviation of `0.301`, which is comparable to the one
    obtained by the proposed method. If the number are indeed cited from
    this paper, additional information about variance needs to be shown
     *and* the citation should be acknowledged in the caption of the
    table, for instance, or in the experimental setup section. It needs
    to be clear to readers which experiments where run by the authors
    and which were cited from another paper.

   - Why is the set of comparison partners changing for the different
    experiments? The DMT method is applicable to 2D and 3D and [code
    appears to be available](https://github.com/HuXiaoling/DMT_loss),
    so it should be part of this comparison, in particular since the
    paper claims that the proposed algorithm outperforms *all* PH-based
    loss functions. (the same comments apply to the papers by Clough et
    al.; for a fair comparison with TDA-based methods, they need to be
    included).

## Detailed comments

- The title is slightly misleading: 'localisation' typically refers to
  the process of finding geometrically useful representations of group
  generators. Here, the filtration incorporates some geometrical
  information but it does not make use of group generators. Using
  'localised' here might be misconstrued at first glance. The title is
  also misleading on a second level because the proposed loss does *not*
  deal with deep learning in general, but with *image segmentation*
  based on deep learning. The loss also does not improve the
  effectiveness of deep learning methods in general. I urge the authors
  to reconsider the title of this paper.

- When discussing persistent homology, consider citing a survey or
  a text book, such as the one by Edelsbrunner and Harer. The work by
  Aktas et al. did *not* describe persistent homology for the first
  time.

- The abstract of the paper is too terse and needs to be slightly
  expanded in order for it to be accessible by non-experts in the field.

- Homology is typically used only in singular form or as a qualifier, as
  in 'homology groups' when discussing TDA literature. The framework
  itself is dubbed 'persistent homology' and a plural use is relatively
  nonstandard.

- The term 'homologies' is incorrectly used throughout the paper; use
  the term 'simplicial chains' or 'topological features' instead. Or,
  for the most precise moniker, you can use 'elements of a homology
  group.'

- Please check carefully when to use 'persistence' and when to use
  'persistent.' The diagram is the 'persistence diagram' because it
  contains information about the 'persistence' of features.

- When claiming that '[...] our technique allows for considerable
  misalignment [...],' provide a reference to the respective section in
  the text.

- The description of persistent homology in Section 3.1 needs to be
  rewritten; use the term 'homology class' if need be instead of
  'homologies.' A more technical description might be helpful here; I am
  not sure whether non-expert readers are able to understand the gist of
  the technique from the current write-up.

- In Equation 2, consider adding a second set of parentheses; the idea
  is that the sum of the two terms is $< s$, correct?

- Figure 3 could be updated to make the difference between the proposed
  filtration function and the 'default' one clearer.

- When discussing experiments, please provide details on how the
  hyperparameters were chosen.

- The *Betti error* needs more details; the present description is
  insufficient.

## Minor comments

- Citations are used inconsistently throughout the paper. Assuming that
  you are using the `natbib` package, use `\citep` for a parenthetical
  citation and `\citet` for a textual citation. Examples of places that
  need such a fix can be found in the paragraph 'Losses that Rely on
  Persistent Homologies' (which I would rename to the singular form, as
  discussed above).



**Summary Of The Paper:**

This paper presents a new approach for detecting topological structures
in 2D and 3D imaging data. Based on the computational topology framework
persistent homology, the paper develops a type of loss term that can be
used to assess to what extent topology is being preserved when solving
segmentation or structural extraction tasks. A suite of experiments is
used to assess the quality of the extracted structures.

**Summary Of The Review:**

This paper deals with an interesting topic, viz. improving 2D and 3D
image tasks by incorporating more prior knowledge about topological
structures. While I consider this to be a highly relevant topic,
I cannot endorse this paper for publication at present, mainly because
of a lack of clarity, a lack of delineation to existing work, and issues
in the experimental setup.

All of these will have to be rectified before I will be able to endorse
the paper for publication; I feel particularly strongly about the
experimental setup, which currently does not enable a fair comparison of
the methods.

I realise that this is not the preferred outcome for the authors; at the
same time, I am confident that with additional rewriting and some
additional work on experiments, this paper can make a strong addition to
the ever-growing body of topology-based machine learning techniques.

**Updated after rebuttal**: I thank the authors for their efforts; while some concerns
where alleviated, I think the updates also showed that the current method does not
substantially improve upon the state of the art in the 2D case, whereas for the 3D case,
additional comparison partners would be required. This is not the only way to improve
the paper, though: another potential direction for future improvement could be to invest
in a more theoretical analysis of the proposed method. If either one of these directions
would be followed, it would serve to substantially improve the quality.

---

> ### Author Response · Authors · 2021-11-19
> **Response to the comments of Reviewer mHrP**
>
> Thank you for the review. We uploaded a new manuscripts and the changes are in blue.
>
>
>
> Q1: several terms are used incorrectly and the underlying concepts are not explained correctly
>
>
>
> A1: Right. We have corrected the errors.
>
>
>
> Q2: missing reference to Clough et al (2020)
>
>
>
> A2: We now discuss it in section 2. It introduces a loss function that relies on the Betti number. For our purposes, this is equivalent to the loss of Hu et al. that compares persistence diagrams, but uses thresholding-based filtration. For ground truth masks, this filtration results in persistence diagrams that are composed of $B$ points, all at $(0,1)$, where $B$ is the Betti number. Minimizing the loss of Hu et al. results in maximizing the persistence of $B$ most persistent topological features and minimizing the persistence of all the remaining ones, just like the loss of Clough et al. (2020). Our approach extends the former but can no longer be interpreted in terms of equalizing the Betti numbers of the prediction and the ground truth, because our filtration results in different topological features than those of unfiltered ground truth.
>
>
>
> Q3: in the experiments, means and standard deviations are only provided for the proposed method
>
>
>
> A3: We added the standard deviations as requested.
>
>
>
> Q4: the numbers for other methods are presumably cited from an additional publication
>
>
>
> A4: No, we ran the experiments reported in Tab 1, except for DMT and Homo-Pre. The authors of the DMT and Homo-Pre papers kindly shared the resulting segmentations with us, which we used to compute all the scores. Importantly, all the experiments in Tab 1 were run using the same network architecture and the same data split as in the DMT and Homo-Pre experiments. The same applies to Tabs 3 and 4. The comparison of Tab 2 is much broader because it contains delineations performed by different architectures. The middle rows contain scores for segmentations the authors shared with us, or available on the Internet. All these networks were trained for best performance: Comparing to results obtained with a training regime selected by the authors to best suit their architecture is a standard practice.
>
>
>
> Q5: The selection of metrics is questionable
>
>
>
> A5: The Reviewer is right that the Dice score and IoU are standard metrics for segmentation. These metrics measure pixel-wise correctness. However, as explained both in the introduction and in section 4.3, for delineating thin curvilinear structures, pixel-wise correctness is a poor performance measure because mislabeling very few pixels often results in disconnecting roads or axons and can introduce false connections between them. In fact, some of our baselines from Tab. 2 (RoadTracer, DeepRoad, SegPath, and PolyMapper) natively produce graphs and it would be awkward to render them for evaluation with the per-pixel scores. Additionally, centerline annotations, used in all our experiments, can never be pixel-precise, which defies pixel-wise evaluation. The metrics we used have been designed to verify topological correctness of delineation results and come from recent first-class road delineation literature (APLS, TLTS, JCT, CCQ), and from the work of Hu et al. (Betti). We used all the metrics measuring topological correctness that we are aware of and we think that, in this sense, the evaluation is very thorough.
>
>
>
> Q6: Request for experiments on 3D experiments on ISBI13, CREMI, 3Dircadb.
>
>
>
> A6: We are focused on delineation of thin, curvilinear structures, like roads and neurites, and our approach consists in regressing the distance to neurite/road center to targets this specific task. The suggested data sets require volumetric segmentation which, for now, is out of the scope of our work. In theory, our loss could be extended to such data by regressing into the distance to region boundary as opposed to the distance to centerline. However, we expect the largest advantage of the proposed loss to be observed for thin curvilinear structures, like the ones we already work with, where disconnections are a frequent problem. To meet the Reviewer's request at least partly, we extended the experimental evaluation with another such data set, called brain. Quantitative results are presented in Tab 4.
>
>
>
> Q7: Was the result of Hu et al. taken from their paper?
>
>
>
> A7: No. The authors shared the predictions with us and we re-computed all the scores.
>
>
>
> Q8: Include 3D DMT and the one by Clough in the experiments.
>
>
>
> A8: Theoretically, DMT could be applied to 3D curvilinear structures, but this has not been demonstrated in the paper. The published evaluation is limited to 3D membranes. Also the publicly available code does not support this. This has been confirmed by the authors in email communication. We are currently trying to make this baseline work, but this requires substantial changes to the code. The method by Clough et al. (2020) is effectively equivalent to the method by Hu et al. (2019) as discussed in A2.

---

> > ### Author Response · Authors · 2021-11-19
> > **Response to detailed comments**
> >
> > We modified the paper according to Reviewer's detailed comments. For hyperparameters please check sections 4.2 and A.2.
> > Concerning mixing citation styles, our intention was to follow the style of the ICLR template that requires parenthetical citation to be used when the author's name is not a part of the sentence and a plain citation when it is.

---

> > ### Comment · Reviewer_mHrP · 2021-11-19
> > **Questions about sdev / minor comments**
> >
> > Thanks for updating the paper. Please clarify a few things for me:
> >
> > > A3: We added the standard deviations as requested.
> >
> > How are standard deviations computed? Over 10 different splits of the data?
> >
> > > Comparing to results obtained with a training regime selected by the authors to best suit their architecture is a standard practice.
> >
> > If I understand you correctly, you are using the models without additional re-training because they are already optimally trained for the task?
> >
> > > A: Our intention was to follow the style of the ICLR template that requires parenthetical citation to be used when the authors name is not a part of the sentence and a plain citation when it is.
> >
> > That's absolutely correct. My point is that you appear to use `\citet` at places where you should use `\citep`. For instance, in the conclusion: "Unlike other powerful approaches Oner et al. (2021)" should become " Unlike other powerful approaches (Oner et al., 2021)."

---

> > > ### Author Response · Authors · 2021-11-19
> > > **Re: Questions about sdev / minor comments**
> > >
> > > Q: How are standard deviations computed? Over 10 different splits of the data?
> > >
> > > A: On the Roads data set we follow the splits used by Hu et al., to make the comparison fair. They used a three-way split. The RoadTracer data set consists of aerial photos of 60 different cities, 45 of which are used as the training set. Here, training is performed once and the standard deviations are computed across 15 different test cities. This is also consistent with previous work in road tracing. On the neurons and brains data sets we introduced a three-way split. This is specified in captions of Tables 1-4.
> > >
> > > Q: If I understand you correctly, you are using the models without additional re-training because they are already optimally trained for the task?
> > >
> > > A: For the results in the middle rows of Tab 2, we did not re-run inference, but used the *outputs* that the authors of the methods shared with us directly, in response to our request, or made available on the Internet. This is a common practice, because the networks were trained by the authors to beat the state of the art. We did ensure the evaluation protocols are aligned.
> > >
> > > Q: That's absolutely correct. My point is that you appear to use \citet at places where you should use \citep. For instance, in the conclusion: "Unlike other powerful approaches Oner et al. (2021)" should become " Unlike other powerful approaches (Oner et al., 2021)."
> > >
> > > A: Thank you for the clarification. We applied the changes.

---

> > > > ### Comment · Reviewer_mHrP · 2021-11-19
> > > > **re: standard deviations**
> > > >
> > > > Some more clarifications: when you say three-way split, do you mean a 3-fold cross-validation procedure? I think it is important to ensure that all experiments are measuring essentially the same type of standard deviation, i.e. some approximation to the generalisation error.
> > > >
> > > > Don't think me to be a pedant here; I genuinely care about PH methods and love to see them succeed, but it needs to be clear how they generalise in contrast to other methods; at the moment, I am not sure what the advantage over the proposed method is over existing methods—can you clarify this? Where do you see that the proposed method has an 'edge' over existing ones?

---

> > > > > ### Author Response · Authors · 2021-11-19
> > > > > **Re: Standard deviations**
> > > > >
> > > > > Q: Some more clarifications: when you say three-way split, do you mean a 3-fold cross-validation procedure?
> > > > >
> > > > >
> > > > > A: Precisely speaking, three random splits of the data were generated, and we trained and tested three times, each time on a different train-test split.
> > > > >
> > > > >
> > > > > Q: What is the advantage of the proposed method over the existing ones?
> > > > >
> > > > >
> > > > > A: In terms of performance, the main advantage is in 3D delineation.
> > > > >
> > > > > Even though in 2D our method is slightly outperformed by ConnLoss, our results still carry an important message.
> > > > > Aligning topological descriptors is a promising approach to training deep nets, especially in tasks like delineation, but
> > > > > currently further research in this direction is discouraged by the recently published, unfavorable comparisons to DMT and ConnLoss.
> > > > > We show that this inferior performance is due to properties of the filtration function that were not reported before.
> > > > > We also propose a new filtration function that makes aligning persistence diagrams outperform all the current 2D methods but one.
> > > > > Even if our 2D result is limited to rehabilitating this approach, as opposed to establishing new state of the art, we think this message is of interest to the ICLR audience.

---

> > > > > > ### Author Response · Authors · 2021-11-20
> > > > > > **Split for the Massachusetts data**
> > > > > >
> > > > > > > A: Precisely speaking, three random splits of the data were generated, and we trained and tested three times, each time on a different train-test split.
> > > > > >
> > > > > > Correction: for the Massachusetts data set, the split is actually alphabetic, and the same as used by Hu et al.

---

> > > > > > ### Comment · Reviewer_mHrP · 2021-11-22
> > > > > > **Experiments/Edge**
> > > > > >
> > > > > > Thanks for addressing this! I think if the main advantage is 3D delineation, I think the experimental setup needs to be revisited. It is important to ensure that a variety of different approaches is being compared to; notice that [Hu et al.](https://arxiv.org/pdf/2103.09992.pdf) also discuss such a comparison (both with topological and geometrical approaches). For the current version of the paper, the focus is too much on the 2D data sets where the proposed method does not really exhibit 'an edge.'
> > > > > >
> > > > > > I know that introducing something new into the community is not easy because there's so many things to compare to, but it's important to make it clear what a proposed method should be able to do in advance. By picking the task at which topological features can excel, you also make it easier to get this work published. As it stands now, the paper does not provide a lot of additional insights into the theory, but also does not have an in-depth comparison for the 3D case (in which it might showcase increased expressivity). This is not meant to sound mean—I genuinely like the approach and would like to see it published; at the same time, I also see that there's still some way to go before this paper is ready for publication in my opinion.

---

### Official Review · Reviewer_ne7r · 2021-11-03

**Correctness:** 4
**Technical Novelty And Significance:** 2
**Empirical Novelty And Significance:** 2
**Recommendation:** 5
**Confidence:** 4

**Main Review:**

The proposed filtration is interesting, and the article is overall well written, but I think that the approach seems a bit incremental for now. There is no real theoretical back-up and the results, while being OKish, do not empirically justify by themselves the approach (to my opinion). Moreover, some parts of the writing could be improved.

In particular, I have the following comments:

1. Section 3.1: "the cost of leaving a homology unmatched set to a constant hyper-parameter": this is not correct, the cost of leaving a point in a persistence diagram unmatched is usually its distance to the diagonal, otherwise the distance is not stable.

2. Equation (2): are the values Y[p] and g(p) always comparable, i.e., on the same scale? If one is much larger than the other, it does not make a lot of sense to do a simple sum between the two. It would be more natural to use a linear combination of the two terms, with the combination coefficients chosen a priori or with cross-validation.

3. Given that the authors aim at using two filtrations at the same time, it feels that persistence bimodules would also be an approach to discuss and compare to, especially since multiparameter persistence can now be computed efficiently, at least when there are no more than two filtrations.

4. Figure 4 (c) and (d) do not have names on their axes.

5. The wording chosen in the article is not great: localization has another specific meaning in Topological Data Analysis, it usually refers to the computation of representative cycles corresponding to homology classes. Hence, the title of the article is confusing and misleading for TDA practitioners.

6. Since the persistent homology transform is known to enjoy some nice injectivity and inverse properties (https://arxiv.org/pdf/1810.10813.pdf), I wonder if such results could also be derived for the proposed loss.

7. Section 4.2: "alpha=0": typo

[Post rebuttal comment]: I updated my review after the author's response.

**Summary Of The Paper:**

In this article, the authors propose a new topological loss for training neural networks on images representing curvilinear structures. In addition to the usual MSE or cross-entropy losses, they suggest to add a regularization term based on persistent homology, in order to preserve the topological properties in the images predicted by the network. However, the common filtrations proposed in the literature (pixel values, distance maps) for doing so have their own flaws and are not always suited for such images, so the authors propose a new filtration that combines those common filtration with filtrations based on heights, i.e., with the persistent homology transform. Using this new filtration, they then provide a few experiments on which their loss compares favorably to other topological and non-topological losses (used on the same fixed neural net architecture).

**Summary Of The Review:**

The proposed approach is definitely promising, but I think that the work is too incremental for now to be accepted as is.

---

> ### Author Response · Authors · 2021-11-19
> **Response to the comments of Reviewer ne7r**
>
> Thank you for the review. We marked the changes to the manuscript in blue. Let us address your remarks.
>
> Q: No real theoretical back-up
>
> A: Indeed, we do not offer a new theoretical result in topology, but we do present an important observation about the existing PH-based loss functions:
>
> 1) When a binary ground-truth is filtered by thresholding, as done by Hu et al. (2019), all resulting persistence intervals are (0,1). In other words, a bar code obtained by thresholding the ground truth encodes no more information than the corresponding Betti number.
>
> 2) Filtering by thresholding a distance map computed for the ground truth represents the topological features of the original, binary ground truth by persistence intervals with the left limit equal zero. This is because, during filtration, these topological features emerge for the lowest threshold. Using only one of the two persistence limits to describe these features clearly limits the expressive power of the persistence diagrams.
>
> Limiting the expressive power of the persistence diagram defeats the purpose of representing the data with a bar code, so the existing PH-based losses do not fully benefit from the power of PH. We believe that anyone interested in using PH to training deep nets should be told about this.
>
> Furthermore, we would like to draw the reviewer's attention to the fact that our new filtration is not just a heuristic, but a combination of the usual filtrations by thresholding images and height functions. This approach is legitimized by the main result of Leo Betthauser's doctoral thesis: a binary image can be reconstructed from four persistence diagrams obtained by using height functions with well chosen directions. Even though our images are greyscale, as opposed to binary, this result justifies the use of multiple filtration directions. Moreover, our approach also relates to multiparameter persistence, as it can be seen as a line in the fibered barcodes defined in (Carriere and Blumberg, NIPS 2020).
>
> We added this discussion to the manuscript.
>
> Q: experiment results are OK, but not good enough to justify the approach.
>
> A: Our real target is 3D where approaches that work well in 2D do not apply anymore. The 2D experiments are there to show that our approach improves upon other PH techniques, not necessarily that this is the best possible technique for road delineation.
>
>
> Q1: The cost of leaving a point in a persistence diagram unmatched is usually its distance to the diagonal, otherwise the distance is not stable
>
> A1: Thank you, we corrected this error.
>
> Q2: Are the values Y[p] and g(p) always comparable?
>
> A2: Both the orientation and the slope of $g(p)$ are selected randomly from the interval [0.1,0.3, as explained in the list at the bottom of page 4. Any slope in this interval give comparable values to distance maps in 64 by 64 windows. Values below 0.1 may not create enough separation between homology classes to obtain correct matching. Values higher than 0.3 may alter the location of critical points. The orientation of $g(p)$ is selected randomly in order to increase the probability of capturing different topological errors.
>
> Q3: Would persistence bimodules would also be an approach to discuss and compare to, especially since multiparameter persistence can now be computed efficiently, at least when there are no more than two filtrations.
>
> A3: Even though there are ways to compute invariant for multiparameter persistence, none of them can currently be combined with deep learning as far as we know. We decided to use combine the greyscale filtration and height function into one filtration, allowing us to base our work on previous work that use differentiability of persistence diagrams.
>
> Q4: Figure 4 (c) and (d) do not have names on their axes.
>
> A4: We added the labels.
>
> Q5: The wording chosen in the article is not great: localization has another specific meaning in Topological Data Analysis, it usually refers to the computation of representative cycles corresponding to homology classes. Hence, the title of the article is confusing and misleading for TDA practitioners.
>
> A5: Thank you for pointing this out, we changed the wording and the title.
>
> Q6: Since the persistent homology transform is known to enjoy some nice injectivity and inverse properties (https://arxiv.org/pdf/1810.10813.pdf), I wonder if such results could also be derived for the proposed loss.
>
> A6: As mentioned in the first comment, it is indeed true that the persistent homology transform completely determines a binary image with only 4 (well chosen) directions. However, this does not work for greyscale images, because a peak could be higher than the “angle” of the slope that tilts the image. We therefore chose not to discuss this aspect. However, we now motivate our approach using both multipersistence and the persistent homology transform.

---

> > ### Author Response · Authors · 2021-11-20
> > **Additional Experiment**
> >
> > Q: Experiment results are OK, but not good enough to justify the approach.
> >
> > A: We extended the experimental evaluation with another 3D data set, called Brain. Quantitative results are presented in Tab 4.

---

> > > ### Comment · Reviewer_ne7r · 2021-11-22
> > > **Response**
> > >
> > > Thank you for the time taken to answer my comments. I like the new experiment, and I like the changes made to the text, so I increased my grade in the review. I still think the paper is a bit too incremental, but I won't oppose acceptance if the other reviewers think it is good enough.

---

> > > > ### Author Response · Authors · 2021-11-22
> > > > **Response**
> > > >
> > > > Thank you for taking our updates into consideration.

---

### Public Comment · ~Yuri_Smirnov1 · 2021-11-15
**Missing citation**

A missing citation. Since you employ the persistence diagrams of complexes, here is the reference where they were first introduced, under the name of canonical forms : Barannikov, S. "The Framed Morse Complex and its Invariants", Advances in Soviet Mathematics, 21: 93–115 (1994). Also, the computation of persistence diagrams is based on the algorithm described in section 2.1 of this reference.

---

> ### Author Response · Authors · 2021-11-19
> **re:Missing citation**
>
> Thank you, we added this reference.

---

### Decision · Program_Chairs · 2022-01-20

**Decision:**

Reject

**Comment:**

The paper proposes a method for segmentation of thin structures in 2D and 3D, based on persistent homology and using a new filtration. The method performs similarly to state-of-the-art methods on 2D datasets and outperforms some baselines in 3D.

After considering the authors' response and discussing, the reviewers have not arrived at a consensus.

Pros include:
- Simple and reasonable approach
- Fairly strong experimental results

Some cons are:
- Missing theoretical contributions
- Experimental results on 2D datasets are not that strong, while on 3D datasets important baselines are missing
- At times unclear/unconventional presentation

Overall, at this point I recommend rejection. The paper is promising, but since the main claim is good performance on 3D data, it is important to have a thorough empirical evaluation there, with the relevant baselines (as mentioned by the reviewers). I very much encourage the authors to polish the paper and submit to a different venue.